# Addressing Natural Killer Cell Dysfunction and Plasticity in Cell-Based Cancer Therapeutics

**DOI:** 10.3390/cancers15061743

**Published:** 2023-03-13

**Authors:** Kassandra M. Coyle, Lindsey G. Hawke, Mark L. Ormiston

**Affiliations:** 1Department of Medicine, Queen’s University, Kingston, ON K7L3N6, Canada; 2Department of Biomedical and Molecular Sciences, Queen’s University, Kingston, ON K7L3N6, Canada

**Keywords:** NK cell dysfunction, NK cell plasticity, TGFβ, HIF1α, indoalemine-2,3-dioxygenase, NK cell therapeutics

## Abstract

**Simple Summary:**

The natural killer (NK) cells of the immune system identify and remove stressed, infected, or cancerous cells in the body. This anti-tumor functionality has been harnessed through promising cell-based therapies that involve the isolation, expansion, activation, and delivery of NK cells for the treatment of several cancers. A variety of techniques have been developed to genetically modify or otherwise improve the activity of these therapeutic NK cells. However, certain elements of the tumor microenvironment are known to alter NK cell functions, suppressing their killing capacity and enhancing their ability to stimulate the formation of blood vessels that support tumor growth. This review summarizes current NK cell-based cancer therapies and discusses improvements that are being pursued to address these mechanisms of tumor-mediated NK cell suppression.

**Abstract:**

Natural killer (NK) cells are cytotoxic group 1 innate lymphoid cells (ILC), known for their role as killers of stressed, cancerous, and virally infected cells. Beyond this cytotoxic function, NK cell subsets can influence broader immune responses through cytokine production and have been linked to central roles in non-immune processes, such as the regulation of vascular remodeling in pregnancy and cancer. Attempts to exploit the anti-tumor functions of NK cells have driven the development of various NK cell-based therapies, which have shown promise in both pre-clinical disease models and early clinical trials. However, certain elements of the tumor microenvironment, such as elevated transforming growth factor (TGF)-β, hypoxia, and indoalemine-2,3-dioxygenase (IDO), are known to suppress NK cell function, potentially limiting the longevity and activity of these approaches. Recent studies have also identified these factors as contributors to NK cell plasticity, defined by the conversion of classical cytotoxic NK cells into poorly cytotoxic, tissue-resident, or ILC1-like phenotypes. This review summarizes the current approaches for NK cell-based cancer therapies and examines the challenges presented by tumor-linked NK cell suppression and plasticity. Ongoing efforts to overcome these challenges are discussed, along with the potential utility of NK cell therapies to applications outside cancer.

## 1. Introduction

In humans, conventional NK cells can be divided into two distinct subsets: highly cytotoxic CD56^dim^CD16^+^ NK cells, which predominate in the peripheral circulation, and CD56^bright^CD16^+/−^ NK cells, which exhibit reduced cytotoxic function, are primarily localized to tissues and secondary lymphoid organs [1,2,3], and are more commonly associated with the regulation of broad immune responses through the secretion of various cytokines (IFNγ, TNFα, GM-CSF, IL-10, IL-5, and IL-13) and chemokines (MIP-1a, MIP-1β, IL-8, and RANTES) [3,4]. CD56^dim^CD16^+^ NK cells make up 90% of peripheral blood and splenic NK cells and exhibit a potent cytolytic activity via the integration of signals from an array of activating and inhibitory receptors [5,6]. In this process, deactivating or inhibitory receptors recognize “self” MHC class-I (MHC-I) molecules that are expressed on the surface of healthy cells and contribute to NK cell self-tolerance. In contrast, virally infected or tumor cells that downregulate surface MHC-I to avoid immune recognition exhibit a “missing self” phenotype, resulting in decreased inhibitory signaling, NK cell degranulation, and lysis [6].

Alternatively, the recognition of antibody-bound target cells by NK cell Fcγ receptors can induce lysis via antibody-dependent cell cytotoxicity (ADCC) [7]. NK cells also express several death ligands with potential cytotoxic activity, such as TNFα, Fas ligand (FasL), and TRAIL. However, only FasL and TRAIL have been shown to induce killing through death receptor-induced apoptosis [8]. In addition to the direct lysis of target cells, the secretion of IFNγ by activated NK cells enhances antiviral, antibacterial, and anti-tumor activity through a variety of mechanisms, including the promotion of macrophage activation, improved dendritic cell antigen presentation, and enhanced lymphocyte endothelium interactions, as well as the regulation of cellular proliferation and apoptosis [9,10,11,12,13,14].

## 2. NK Cell Therapies

### 2.1. Primary NK Cell Sources

To date, therapies that aim to harness the anti-tumor function of NK cells have involved either the adoptive transfer of ex vivo expanded NK cells, with or without genetic modification [15,16,17,18], or the promotion of in vivo NK cell activation through the administration of recombinant proteins, such as interleukin-15 (IL-15) superagonists [19,20]. For cell-based approaches, several sources have been explored, including primary NK cells from autologous or allogeneic peripheral blood, apheresis products, or umbilical cord blood (UCB) (Figure 1). Primary NK cells comprise only 10% of all lymphocytes in peripheral blood, limiting the number of cells that can be harvested for therapeutic applications and necessitating extensive purification methods, such as the magnetic depletion of undesired cell types [21]. NK cells are comparatively enriched in UCB, comprising 15–30% of total lymphocytes, but are immature and may be less cytotoxic compared to other sources [22,23].

Alternatively, NK cells can be safely generated from CD34^+^ hematopoietic progenitors, induced pluripotent stem cells (iPSC), or embryonic stem cells [24,25]. While iPSC-derived NK cells offer a potentially infinite source of homogenously differentiated NK cells and can be functionally enhanced by genetic modification with relative ease, few groups have successfully created functional NK cells from this cellular source [26]. Woan and colleagues reported the use of a clonal iPSC-NK cell line, engineered to express both a high-affinity, non-cleavable version of the CD16a Fc receptor and a membrane-bound IL-15/IL-15 receptor-α fusion protein [27]. These cells demonstrated potent anti-tumor activity in vitro, as well as in myeloma and acute myeloid leukemia (AML) xenograft models. Clinical trials are currently underway [28]. The use of UCB-derived CD34^+^ cells as a source for NK cell therapies is also being explored by Deverra Therapeutics, through their non-engineered product DVX201 [29].

### 2.2. Immortalized NK Cell Lines

Immortalized NK cell lines also offer an alternative to primary NK cells, as they can be cultured indefinitely and are amenable to genetic manipulation. Several NK cell lines have been explored for therapeutic use, including NKL, NKG, NK-YS, YT, YTS, NK-92 cells, and high-affinity NK cells (haNKs). Originally isolated from an individual with non-Hodgkin’s lymphoma, NK-92 cells display high levels of anti-tumor cytotoxicity more consistently and reproducibly than other cell lines [30,31], are easily expandable to the clinical grade levels required for treatment [32], and have been used in recent clinical trials in patients with advanced, treatment-refractory malignancies [33,34]. HaNKs are NK-92 cells that have been engineered to express a high-affinity CD16 receptor for increased ADCC and endogenous production of IL-2 to maintain cytotoxic function [35,36]. While advantageous, the clinical application of these cells is not without limitation. NK-92 cells must be irradiated to protect from malignant expansion, potentially limiting their in vivo efficacy and persistence.

### 2.3. NK Cell Expansion and Activation

Regardless of their source, NK cells must be expanded to generate sufficient numbers of highly functional cells for therapeutic applications (Figure 1). Clinical scale expansion can be achieved using mitotically inactivated feeder cells, such as peripheral blood mononuclear cells (PBMCs), K562 or Jurkat cells, with or without genetic modification [37,38,39]. Relative to freshly isolated NK cells, expanded populations adopt an activated phenotype with increased expression of activating receptors NKG2D, NKG2C, NKp30, NKp44, DNAM-1 and key effector molecules, such as TRAIL, FasL and granzymes [40].

NK cells have also been successfully expanded in feeder-free systems using cell culture flasks, bags, or bioreactors in the presence of high doses of cytokines, such as IL-15 and IL-21 [41,42], cytokine-conjugated magnetic beads [43,44], or plasma membrane-derived particles [45]. However, these methods typically yield lower cell numbers than what is achieved with feeder cell-based expansion. More recently, advanced systems involving dissolvable polymer-based microspheres [46] or streamlined expansion protocols [47] have allowed for improved yields and the incorporation of nonviral genome editing into feeder-free expansion systems.

Although growing evidence supports the safety and efficacy of the majority of these expansion techniques [48], ex vivo NK cell expansion introduces the potential for senescence and exhaustion [49], resulting in the need for expanded NK cells to be activated or “primed” with cytokines like IL-2, IL-12, IL-15, and IL-18 to achieve maximal efficacy on tumors [50,51]. These cytokines, along with IL-21 and type I interferons, are central to the maturation, activation and survival of NK cells. IL-2 is critical to NK activation, can rejuvenate exhausted NK cells, and restores or preserves their cytotoxic potential in response to various stressors or following exposure to multiple myeloma [52,53,54]. Similarly, the stimulation of NK cells with various combinations of IL-2, IL-12, IL-15 and IL-18 increases the production of critical effector cytokines like IFNγ, IL-8, and TNF-α [55], enhances the responsiveness of the cells to the integrin, LFA-1, and improves subsequent receptor stimulation [56].

### 2.4. Alternative Approaches for NK Cell Activation

In addition to cytokine-based approaches, genetic manipulations, such as transgenic expression of the Chimeric Antigen Receptor (CAR), have also been explored as a means to enhance NK cell activation for therapeutic applications (Figure 1) [57]. CAR is a recombinant protein that can be introduced into cytotoxic lymphocytes to enable tumor antigen recognition and trigger activation [58]. CAR expression allows NK cells to specifically target cancer cells through recognition of tumor associated antigens. The generation of CAR-NK cells involves transducing isolated NK cells with CAR-encoding genes and expanding these cells prior to adoptive transfer into the patient [59]. Recently, UCB-derived CAR-NK cells were used to treat patients with relapsed or refractory CD19^+^ cancers, resulting in complete remission in the majority of patients without significant toxic effects [60]. Multiple clinical trials are currently underway [61], and ongoing studies are investigating the use of CAR-NK cells for the treatment of non-malignant targets, such as HIV-infected cells [62,63].

Despite these advantages, the complex CAR-NK cell manufacturing process is hindered by inefficient transduction methods [64], as well as the poor in vivo persistence shared by many NK cell therapy platforms. The personalized approach required for CAR-NK cells is also extremely expensive, time consuming, and difficult to apply large-scale. These limitations, which are not exclusive to CAR-NK approaches (Table 1), demonstrate a clear need for effective “off-the-shelf” therapies. One such approach is bi- and tri-specific antibodies (BiKEs and TriKEs), which trigger ADCC by binding both CD16 on NK cells and specific tumor antigens, creating a connection-like bridge between these cells and allowing for NK cell activation [65]. TriKEs have also been modified to incorporate cytokines like IL-15 to increase in vivo persistence and activation of NK cells [66]. In vitro, TriKEs have demonstrated enhanced NK cell activation and killing of both AML cell lines and patient-derived AML blasts [67]. A clinical trial in the treatment of high-risk hematological malignancies is currently underway [68].

The NK cell engager platform (ANKET) has also been investigated as a means to harness NK cells as next-generation cancer immunotherapies. These NK cell engagers link a monoclonal antibody targeting the activating NK cell receptor NKp46 (or NKp30), an Fc fragment to promote ADCC via CD16, and an antibody targeting a tumor associated antigen, to enable the tumor-localized activation of host NK cells [76]. Work with these engagers has shown promise in targeting various cancer cell lines [76,77,78]. Recently, Demaria and colleagues reported a tetraspecific CD20-ANKET, which engages NKp46, CD16a, the beta chain of the IL-2 receptor, and a tumor associated antigen to induce preferential NK cell activation and target cell killing [79]. This tetraspecific CD20-ANKET induced the local control of tumors in non-human primates, without significant adverse side effects [79].

## 3. NK Cell Impairment in Cancer

Even though extensive progress has been made towards the efficient isolation, expansion and functional enhancement of NK cells for therapeutic applications, these approaches are still limited by disease-associated environmental challenges that suppress or modify NK cell function in vivo. NK cell dysfunction is a major component of the development, progression, and survival of many cancer models and is strongly correlated with worsened survival outcomes in patients [80,81]. A decrease in circulating NK cells is often accompanied by poor infiltration of NK cells into the tumor microenvironment [3,82,83,84,85]. NK cells isolated from tumors also produce less IFNγ, CD107a, granzyme B, and perforin when compared with NK cells from peritumor regions or peripheral blood [83,85,86] and exhibit both a downregulation of NK activating receptors, including NKG2D, CD16, NCRs (NKp30, NKp44 and NKp46), CD226, and 2B4 [83,87,88,89], and elevated inhibitory receptors, such as NKG2A [86].

These functional impairments have been attributed to several cellular processes, including exhaustion, where persistent NK activation results in decreased effector functions and poor control of malignancies and infections [90], as well as anergy, which is defined as a hypo-reactive state that is attributed to either excessive inhibitory signaling [91] or NK activation in the absence of proper licensing via inhibitory receptor binding [92,93]. NK cell dysfunction can also arise from the deprivation of survival signaling from cytokines like IL-15 [94], or as a result of suppressive signals in the tumor microenvironment, such as TGFβ, hypoxia and IDO, which are discussed in detail below (Figure 2). In addition to the more classical forms of NK functional impairment, recent work has also identified substantial potential for plasticity between ILC1 subsets, through which tumor-associated factors can drive the conversion of conventional cytotoxic NK cells towards tissue-resident or ILC1-like phenotypes that are either poorly cytotoxic or may even aid in tumor growth via the promotion of enhanced tumor vascularization [95,96,97,98,99]. Together, these mechanisms of impairment demonstrate how the tumor microenvironment can reduce the efficacy of conventional NK cell-based therapeutics and highlight the need to both understand and specifically target disease-mediated NK cell suppression to increase the performance of next-generation therapeutic approaches.

### NK Cell Plasticity

Since the first identification of distinct ILC populations, group 1 ILCs have been classified into two subgroups: conventional NK cells, which express cytotoxic factors like IFNγ, perforin, and granzyme B, as well as the transcription factors T-bet and Eomes, and ILC1 cells, which are Eomes-negative, poorly cytotoxic and largely reside in non-lymphoid tissues, such as the skin, liver and uterus [100,101]. Although these subsets were originally believed to be the developmentally distinct progeny of discrete lymphoid precursors, later work identified the potential for factors such as TGFβ to induce the conversion of circulating NK cells into intermediate ILC1 or ILC1-like phenotypes, characterized by a reduction in cytotoxic capacity, the acquisition of ILC1-linked surface markers, and, for ILC1-like cells, a reduction in Eomes expression [102]. Importantly, converted ILC1s and intermediate ILC1s are unable to control tumor growth and metastasis, providing a direct mechanism by which this plasticity can facilitate the escape of tumors from NK cell-mediated immunosurveillance. This process of conversion is not dependent upon canonical TGFβ signaling via the Smad family of signal transducers [103], but instead relies on non-canonical TAK1-mediated activation of p38 MAP Kinase via a process that is synergistically enhanced by IL-15 [104]. The spontaneous conversion of NK cell to an ILC1-like phenotype has been demonstrated in mice harboring a conditional deletion of the *Smad4* gene in their Group 1 ILCs [103], confirming that canonical signaling via Smad4 actually works to prevent conversion via non-canonical TGFβ pathways. Importantly, these *Smad4*-deficient mice are also unable to control tumor metastasis or viral infection, highlighting the link between NK plasticity and cancer.

## 4. TGFβ-Mediated NK Cell Impairment

Beyond its specific actions on NK cell plasticity, TGFβ is a major immunosuppressive cytokine that correlates with poor prognosis and reduced NK cell activity in multiple cancer types [105,106]. In pancreatic cancer, membrane-bound TGFβ_1_ on cancer-expanded myeloid-derived suppressor cells (MDSC) was found to induce NK cell dysfunction through diminished NKG2D expression and impaired NK cytotoxicity [107,108]. TGFβ downregulates the expression of NKG2D on NK cells and CD8^+^ T cells in several cancer models [105,106,107,109], and has been shown to suppress IL-15 and STAT5-mediated NK cell activation through the blockade of mTOR activation [110]. Importantly, NK cell-specific deletion of the TGFβ type II receptor (TGFβRII) causes enhanced anti-tumor activity in multiple mouse models of metastasis, whereas constitutive TGFβ signaling arrests NK cell development and drives increased tumor growth [110]. These effects of TGFβ were linked to a direct inhibition of mTOR-dependent metabolic activity in NK cells stimulated by IL-15, and point to the stimulation of NK cell metabolic activity as a potential strategy to promote or enhance NK cell-based cancer therapies.

As detailed above, TGFβ is also known to impair the cytotoxic function of NK cells by inducing the conversion of cytotoxic circulating NK cells to a poorly cytotoxic tissue-resident or ILC1-like cell type [104,111]. In mice, CD49a^+^CD49b^−^ tissue-resident NK cells in tissues such as the liver, skin and uterus can be distinguished from circulating NK cells, which are CD49a^−^CD49b^+^. Uterine NK (uNK) cells are among the best-defined of these tissue-resident NK cell subsets [112,113]. During the initial stages of pregnancy, uNKs increase in number at the site of embryo implantation [114,115,116,117,118] to accompany extensive uterine vascular remodelling that allows for a 10-fold increase in blood supply to the intervillous space [119]. In pregnancy, uNK-derived IFNγ, matrix metalloproteinases (MMPs), and angiogenic growth factors such as vascular endothelial growth factor (VEGF), placental growth factor (PIGF) and angiopoietin (ANG) 1 and 2, contribute to uterine vascular remodeling and immunoregulation during normal pregnancy [112,120].

In addition to TGFβ, which can drive the conversion of circulating cytotoxic NK cells to poorly cytotoxic ILC1-like cells with impaired anti-tumor function and tissue-resident surface marker expression [103,121], exposing circulating NK cells to conditioned medium from decidual stromal cells or a combination of TGFβ, hypoxia, IDO, and demethylating agents, has also been shown to promote the acquisition of uNK phenotypic and functional attributes by peripheral blood NK cells [104,122,123,124]. These attributes include reduced cytotoxicity and the expression of adhesion molecules, such as CD9 and CD103, which are directly linked to tissue residency. While several studies have associated this conversion with the enhanced production of VEGF [125] and have shown that NK-derived VEGF is a critical regulator of tumor vascularization and growth in mouse models [126], in vitro studies examining the discrete effects of TGFβ and hypoxia on cultured human peripheral NK cells have shown that VEGF production is largely hypoxia dependent and is not enhanced by the process of TGFβ-mediated NK cell conversion [124]. These findings do leave open the possibility of a two-stage model, whereby TGFβ promotes the movement of NK cells into tissues, and the resultant reduction in oxygen tension drives VEGF production (Figure 2A).

### Current Work Targeting the TGFβ-NK Cell Axis

Multiple studies have targeted TGFβ-mediated immunosuppression in cancer through the use of TGFβ ligand traps or receptor kinase inhibitors [127,128,129,130,131,132]. While many of these works demonstrated beneficial effects via the preservation of NK cell function [133], these strategies, which are based on the global blockade of TGFβ signaling, are unlikely to translate into effective therapies due to unwanted side-effects, including autoimmune inflammation and cardiovascular toxicity [134,135]. As an alternative, more targeted approaches that directly address the impact of TGFβ on NK cell phenotype and function are currently being explored and have shown promise in early pre-clinical studies (Figure 3A). UCB-derived NK cells, modified to express a dominant-negative TGFβ receptor that is coupled to an NK-specific activating signal, demonstrated higher cytotoxic activity in TGFβ-rich environments, both in vitro and in vivo [136]. These TGFβ receptor-modified NK cells are phenotypically and functionally similar to unmodified cells, with an added protection against exogenous TGFβ in a xenograft model of neuroblastoma. Similarly, the genetic modification of NK-92 cells to express a chimeric receptor with a TGFβRII extracellular and transmembrane domain, linked to an NK-activating NKG2D intracellular domain, exhibited resistance to TGFβ, a higher killing capacity, and elevated IFNγ production, as well as improved inhibition of tumor growth in vivo [137]. NK-92 cells modified to express a dominant negative TGFβRII also demonstrated insensitivity to TGFβ-mediated suppression in vitro and a capacity to decrease tumor proliferation, reduce lung metastasis, and enhance the survival of mice in a lung cancer model [138].

## 5. Hypoxia-Mediated NK Cell Impairment

In certain tumors, the absence of a functional circulation can give rise to a hypoxic microenvironment, altering the function of multiple immune cell types and favoring disease progression (Figure 2B). NK cells are sensitive to hypoxia, as their cytolytic function is impaired and expression of the activating receptors, NKp46, NKp30, NKp44 and NKG2D, is reduced [139,140,141,142]. In response to hypoxia, NK cells upregulate hypoxia-inducible factor 1α (HIF-1α), a transcription factor that regulates oxidative and glycolytic cellular metabolism, increasing anaerobic metabolism, decreasing mitochondrial oxygen consumption, and driving alterations in the NK cell transcriptome that govern the adaptation to oxygen-depleted environments [143,144]. IL-2 primed NK cells subjected to short-term (16 h) and prolonged (96 h) hypoxia are functionally reprogrammed, exhibiting differential expression of a large number of genes, including proinflammatory cytokines, chemokines, and chemokine-receptors [145]. These changes result in increased NK cell migration and decreased cytotoxicity [146], as well as reduced secretion of IFNγ, TNF-α, GM-CSF, and members of the CC chemokine family (CC3 and CC5) [145].

Additionally, increased HIF-1α expression by tumor cells can decrease NK cell-mediated killing by downregulating tumor-derived MHC class I chain-related genes [147] or by promoting the shedding of MICA, a ligand that triggers the cytolytic action of immune effectors, from the surface of tumor cells [148,149]. This HIF-1α-mediated NK cell evasion is thought to be linked to increased expression of the metalloproteinase ADAM10, which is required for the hypoxia-induced shedding of MICA [150].

As detailed above, exposure to hypoxia can promote NK cell-mediated angiogenesis via HIF-1α-induced VEGF production [123,145]. Targeting this process may have therapeutic potential, as mice bearing an NK-specific deletion of VEGF or HIF-1α exhibit reduced growth of solid tumors and impaired tumor vascularization [151,152]. Of note, tumors from mice lacking HIF-1α in their NK cells exhibited high-density networks of immature, non-functional vessels that were attributed to the reduced infiltration of sVEGFR1-expressing NK cells into hypoxic regions of the tumor, increased VEGF bioavailability, and non-productive angiogenesis [152].

### Current Work Targeting Hypoxic NK Cells

Inhibition of HIF-1α with the small molecule KC7F2 has been explored as a potential avenue for cancer therapy. Single-cell RNA sequencing of tumor infiltrating NK cells revealed that HIF-1α inhibition maintained the expression of activation markers and effector molecules, while preserving cytotoxic activity in hypoxic conditions [153]. Moreover, the conditional deletion of HIF-1α in NK cells decreased tumor burden and enhanced survival in several cancer models, including RMA-S lymphoma, Lewis lung carcinoma and B16-Rae1 melanoma [153].

Downregulation of HIF-1a signaling within the tumor itself can also allow for enhanced immune activity. Intratumoral gene transfer of an antisense HIF-1α plasmid has been shown to downregulate VEGF and decrease tumor microvessel density in an EL-4 mouse lymphoma model, resulting in the complete and permanent NK cell rejection of small tumors (0.1 cm in diameter) [154]. When synergized with T cell co-stimulatory B7-1-mediated immunotherapy, this antisense HIF-1α therapy caused the NK cell-dependent rejection of larger EL-4 tumors up to 0.4 cm in diameter that were refractory to other monotherapies. Further, mice exposed to this combination therapy resisted a rechallenge with parental tumor cells, indicating systemic anti-tumor immunity [154].

Beyond the direct targeting of HIF-1α, IL-2 activation can also restore the killing potential of NK cells that have been exposed to hypoxia, allowing for maintained degranulation in response to target multiple myeloma cells, despite hypoxic conditions [54]. Although off-target effects hinder clinical feasibility of systemic recombinant IL-2 treatment, targeted approaches, such as haNK cells, which express a high-affinity CD16 receptor and internal IL-2, offer the potential of sustained killing under hypoxic conditions, when compared to healthy donor NK cells whose cytolytic abilities are impaired (Figure 3B) [155].

## 6. IDO-Mediated NK Cell Impairment

IDO is an intracellular enzyme that catabolizes tryptophan to N-formylkynurenine, which is then converted into extracellular messengers that are collectively known as kynurenines [156]. In addition to its broad immunosuppressive activity in autoimmune diseases [157,158,159], chronic infections [160], and cancer [161], IDO-derived kynurenine exposure induces NK cell apoptosis through a reactive oxygen species-mediated pathway and impairs NK cell function by suppressing surface expression of the NK activating receptors NKp46 and NKG2D (Figure 2C) [162,163,164]. Short hairpin RNA silencing of IDO in ovarian cancer cells has been shown to reinforce cancer cell sensitivity to NK cells in vitro, while also reducing tumor growth, decreasing peritoneal dissemination, and promoting NK cell accumulation in the tumor stroma in vivo [165]. Mechanistic studies examining the impairment of NK cells by IDO have shown that high IDO expression significantly reduces NK cell cytotoxicity via the miR-18a/NKG2D/NKGD2L regulatory axis, providing for a useful target for specific future manipulations [166].

### Current Work Targeting the IDO-NK Cell Pathway

Based on its influence on NK cells, as well as other immune cells, IDO inhibition has arisen as a potential target for anti-cancer therapeutics. Several IDO inhibitors are being tested in both pre-clinical and clinical trials (Figure 3C) [167]. Most inhibitors bind the IDO enzyme, preventing the conversion of tryptophan into the kynurenines that enact toxic effects on NK cells and other immune substrates. Global IDO inhibition using the competitive IDO inhibitor, 1-methyl-L-tryptophan (1MT) slowed tumor growth and enhanced tumor rejection in mice injected with IDO-expressing P815B cells [161]. IDO inhibition by 1MT also cooperates with several chemotherapeutic agents to effectively promote regression of established breast cancer tumors that were previously resistant to chemotherapy, suggesting that, while minimally effective on its own, 1MT intervention may be best used in combination with other anti-cancer therapies [168]. In a breast cancer model, the direct administration of 1MT directly into tumor grafts slowed tumor outgrowth, whereas the combination of 1MT with paclitaxel resulted in a 30% decrease in tumor volume within 2 weeks of initiating therapy [168]. Similarly, combining 1MT with the nicotinamide phosphoribosyl transferase (NAMPT) inhibitor, APO866, resulted in a greater effect on both murine gastric and bladder tumor models than either treatment alone [169].

Despite these successes, there is also significant controversy regarding the efficacy of these IDO inhibitors. Many of the current inhibitors have low in vitro activity. 1MT and Norharmane inhibit IDO on micromolar levels, while Epacadostat and Navoximod exert inhibitory effects in the nanomolar range [170]. As such, the in vivo efficacy of these inhibitors has been brought into question. 1MT also exists as two stereoisomers, with potentially different biological properties. It is currently unclear which isomer might be preferable for therapeutic development [171,172]. Beyond these limitations, there is also limited data available on the direct actions of these therapeutics on NK cell function, both in vitro and in vivo.

## 7. Conclusions and Future Perspectives

The NK cell is a highly dynamic member of the immune system that can adapt its phenotype in response to microenvironmental conditions that vary between the circulation and healthy or diseased tissues. Extracellular influences, such as TGFβ and hypoxia, can promote the phenotypic shift in NK cells to a more tissue-resident or ILC1-like phenotype, diminishing their powerful cytotoxic functions in favor of a cytokine-producing cell type with the potential to enhance tumor angiogenesis and growth. While the recent success of CAR-T cell therapies highlights the feasibility of cell-based approaches for enhancing anti-tumor immunity in human subjects, these mechanisms of in vivo suppression represent potentially major limitations on the persistence, efficacy, and, ultimately, the clinical translatability of NK cell-based therapeutics for the treatment of cancer. Ongoing pre-clinical studies exploring the specific targeting of these mechanisms of NK cell suppression, either via the genetic modification of primary or immortalized NK cells, the systemic administration of therapeutic agents, or the targeting of shared mechanisms, such as reduced NK cell bioenergetic metabolism, not only show promise for the treatment of individual cancers, but may one day benefit patients with other diseases linked to NK cell impairment.

NK cell-based therapies are already being investigated [54,155,156,157] in the treatment of HIV [173,174]. Additionally, conditions such as endometriosis and pulmonary arterial hypertension (PAH), a disease of obstructive remodeling in the pulmonary circulation, are also linked to NK cell impairment associated with excessive TGFβ, IDO and HIF activity. It is therefore possible that future NK-cell-based therapies with improved selectivity and efficacy may eventually be explored for these conditions. In endometriosis, crosstalk between endometrial stromal cells and macrophages, and their secretion of IL-10/TGFβ impairs NK cell cytotoxicity, viability, and phenotype via a process that can be partially reversed by anti-TGFβ therapy [175]. Peritoneal fluid analysis from women with endometriosis also identified high levels of IDO, which reduced expression of the activating receptors NKp46 and NKG2D [176].

In PAH, NK cell dysfunction is a feature of both human patients and animal models of disease [177]. NK cells from PAH patients exhibit excessive TGFβ signaling, linked to the reduced expression of activating and inhibitory receptors. Interestingly, pulmonary hypertension develops spontaneously in NK cell-deficient mice [178], suggesting a potentially causal role for NK cells in the development of this disease. Together, these findings point to a future for NK cell-based therapeutics that may extend beyond cancer. With the recent initiation of multiple ongoing clinical trials, and the rise of advanced methods for the expansion, activation, and genetic modification of NK cells for therapeutic applications, the next decade of NK cell therapies offers great promise for translational advances across a wide range of applications.

## Figures and Tables

**Figure 1 cancers-15-01743-f001:**
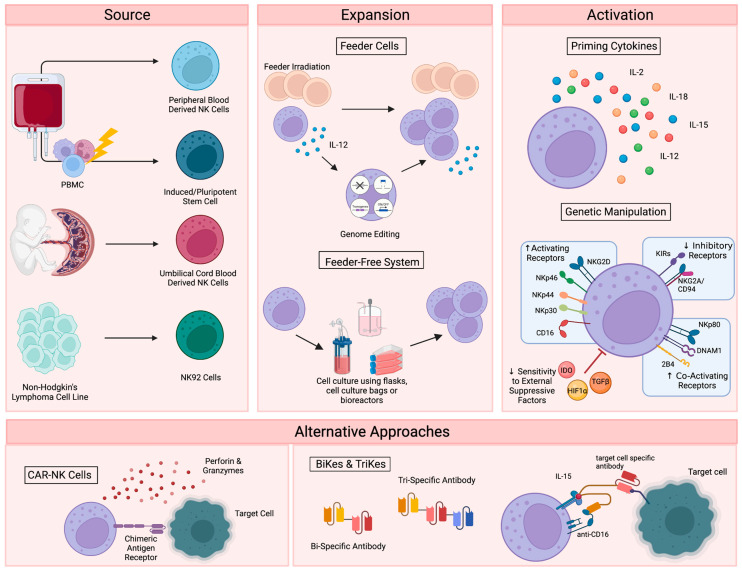
Current NK cell therapies. Schematic showing the current use of NK cell therapies, starting with source, expansion, and activation, then demonstrating the recently developed alternative approaches. Figure created with BioRender.

**Figure 2 cancers-15-01743-f002:**
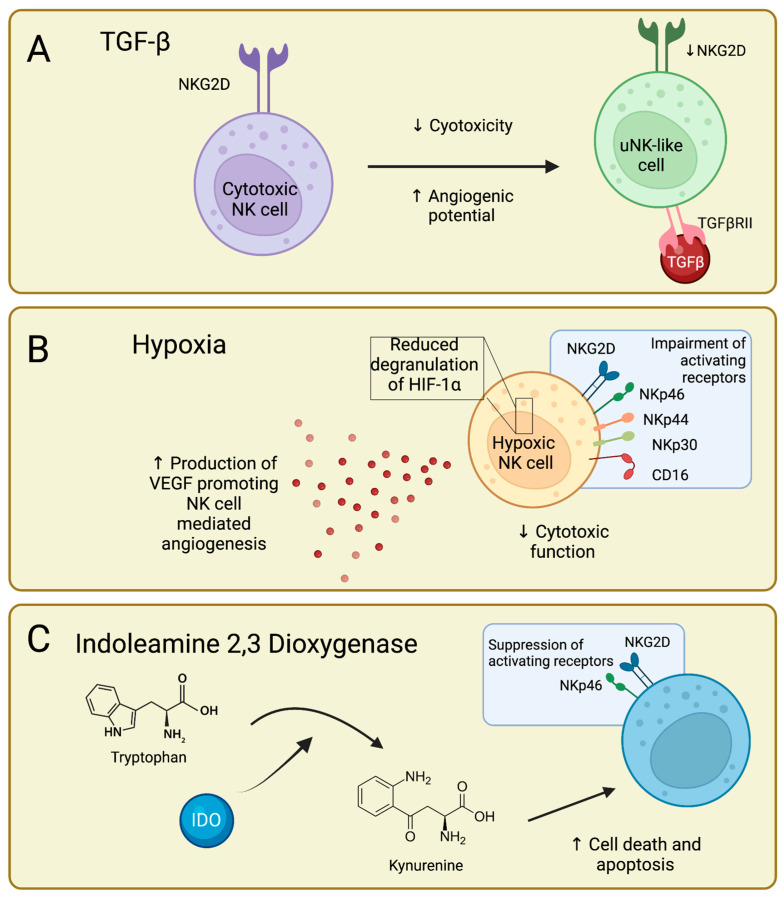
NK cell suppression. Schematic showing NK cell suppression after exposure to (**A**) TGFβ, (**B**) hypoxia, or (**C**) IDO. Figure created with BioRender (https://www.biorender.com/ [accessed on 10 February 2013]).

**Figure 3 cancers-15-01743-f003:**
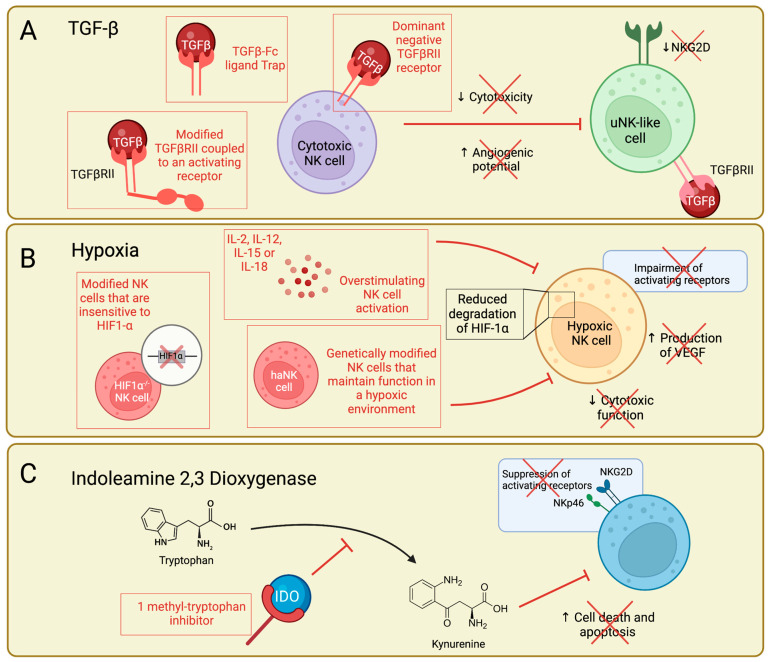
Current interventions targeting NK cell suppressive factors. NK cell suppression via (**A**) TGFβ, (**B**) hypoxia, or (**C**) IDO. Figure created with BioRender (https://www.biorender.com/ [accessed on 11 February 2013]).

**Table 1 cancers-15-01743-t001:** Advantages and Limitations to Current NK Cell Therapies.

	Advantages	Limitations
Source		
Peripheral Blood NK Cells	Reliable source of CD34 progenitor cells [64]	NK cells make up only ~10% of all lymphocytes in peripheral blood
	High expression of CD16+	Extensive purification is required to reduce contamination [21]
	Clinical studies have shown success with these cells after extensive enrichment and purification [69]	Isolating large amounts of PB NK cells is difficult [70,71,72]
		Cryopreservation has been shown to reduce cytotoxicity [71,73]
Umbilical Cord NK Cells	Greater abundance than PB NK cells (15–30% of total lymphocytes) [23]	UCB NK cells are immature
	Fewer contaminating T cells in UCB than PB, reducing the risk of graft-versus-host disease [64]	
	Associated with good tolerance	May have reduced cytotoxic function [22]
	Minimal graft-vs-host-disease or toxicity [24]	
Induced Pluripotent NK Cell	Easily genetically modified	Limited clinical success to date
	High availability	Complex differentiation steps
	Ability to generate multiple doses from a single healthy donor [31,71]	Safety concerns regarding toxicity
Commercial NK Cell Lines	Easy to obtain	Must undergo irradiation to prevent malignant expansion, which could limit persistence.
	Highly cytotoxic	
	Easily expandable [32]	
	NK92 cells are the only cell line that has shown success in pre-clinical studies [31]	Efficiency of cells after expansion is variable (4–95%) [74]
Expansion		
Feeder Cells	Effective expansion of large numbers of NK cells [75].	Difficult to maintain cytotoxic function after expansion [49].
Feeder-Free Expansion	Large amounts of highly active NK cells have been produced	Cytotoxic function after expansion has not been well reported
Activation		
IL-2	Ability to restore NK cell cytotoxicity after exposure to various stressors [54].	Systemic IL-2 leads to significant toxicity
Other Activating Cytokines	Less toxic than IL-2	Thought to provide only minimal clinical benefit
		Many combination therapies are required to provide a therapeutic benefit
Genetic Manipulation	Ability to target specific pathways of interest	Relatively newer area of study
	Ability to avoid toxic effects associated with global therapies	

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
