# Peer review of "Addressing Natural Killer Cell Dysfunction and Plasticity in Cell-Based Cancer Therapeutics"

_cancers, 2023, doi:10.3390/cancers15061743_

Round 1

Reviewer 1 Report

In this paper Coyle et al review the current knowledge on the approaches for NK cell-based anti-tumor therapies. The authors discuss how the tumor microenvironment can reduce the efficacy of conventional NK cell-based therapeutics and highlight the need to specifically target disease-mediated NK cell suppression to increase the performance of next-generation therapeutic approaches.

In my opinion, this is an important topic. Overall, the article is well-organized, well written, and interesting.

I only have some, yet important, comments/suggestions which are listed below.

1. Table1 summarizes the advantages and limitations of NK cell therapies:

- The headings (Source, Expansion, Activation) should be highlighted better (written in bold?, placed in the middle of the two columns?). The two columns (advantages and limitations) should be separated more (possibly by adding vertical line?).

- UCB: “generally, less contaminated” What does it mean? Does it mean that UCB are never contaminated with EBV or CMV? It should be explained better, at least in the text.

- Feeder cells: “Studies have shown effective expansion…” The first part of the sentence (Studies have shown) is not necessary and can be deleted. There is also the number 1 after the ref 177 to be deleted.

2. I suggest making some minor changes to the figures that already provide a clear explanation of the topic discussed:

- Figure 1 TriKes, as reported in lines 157-158, “have also been modified to incorporate cytokines like IL-15”. It is better to represent the cytokine differently from the antibody fragments.

- Figures 2 and 3 are not mentioned in the text. Please add the cross-reference in the text.

- Figure 3: I suggest moving the panels to follow the order of the different paragraphs (TGF-beta, hypoxia, and IDO). The addition of a letter to indicate the different panels allows to mention of Figure 3 and the appropriate panel in the text.

3. A good summary of the current and updated literature is provided even though some references are lacking, not fully appropriate, or incorrectly cited: 

- Line 44: Reference 5 (Roda, D.; Jimenez, B.; Banerji, U. Are Doses and Schedules of Small-Molecule Targeted Anticancer Drugs Recommended by Phase I Studies Realistic? Clinical Cancer Research 2016) is not adequate for cytokine and chemokine production by NK cells.

- Line 53: it would be better to include a more general reference on ADCC by NK cells (and other NK cell functions); for instance, Vivier E. et al. Nat Immunol. 2008.

- Line 71: there are different papers on IL-15 superagonist ALT803 (or N803) more suitable than ref 21. One is “First-in-human phase 1 clinical study of the IL-15 superagonist complex ALT-803 to treat relapse after transplantation by Romee R. et al Blood, 2008”, but there are also many others.

- Line 147: references 63 and 64 concern CAR-T cells. Please provide adequate references for “…the use of CAR-NK cells for the treatment of non-malignant targets, such as HIV-infected cells.” In addition, in the context of CAR-NKs, the work of K. Rezvani's group is extremely relevant and should be mentioned (for example Liu et al. N. Engl. J. Med., 2020).

- Lines 357-358: The sentence “In addition to HIV (?)…” is not clear. References 55, 143-145 are not appropriate.

The whole manuscript should be carefully checked for missing or inappropriate citations.

4. Paragraph 2.4: In addition to BiKes and TriKes the NK cell engager platform (ANKET), which represents a technological platform for harnessing NK cells as next-generation cancer immunotherapies, should also be mentioned and appropriate references included. These NK cell engager molecules trigger NK cell-activating receptors as NKp46 (or NKp30) together with CD16a and target a tumor-associated antigen (TAA) (Gauthier L. et al. Cell, 2019; Colomar-Carando et al. Cancer Immunol Res, 2022; Gauthier L et al. Nat. Biotechnol. 2023). A tetraspecific CD20-ANKETs (IL-2v/aNKp46/Fc/aCD20) which also contains an IL-2v peptide stimulating IL-2R without CD25 involvement has been also produced (De Maria O. et al. Cell Reports Medicine, 2022) and can be mentioned.

Author Response

Response to Reviewer 1

In this paper Coyle et al review the current knowledge on the approaches for NK cell-based anti-tumor therapies. The authors discuss how the tumor microenvironment can reduce the efficacy of conventional NK cell-based therapeutics and highlight the need to specifically target disease-mediated NK cell suppression to increase the performance of next-generation therapeutic approaches.

In my opinion, this is an important topic. Overall, the article is well-organized, well written, and interesting.

We thank the reviewer for their positive assessment of our manuscript.

I only have some, yet important, comments/suggestions which are listed below.

C1) Table1 summarizes the advantages and limitations of NK cell therapies:

C1a) The headings (Source, Expansion, Activation) should be highlighted better (written in bold?, placed in the middle of the two columns?). The two columns (advantages and limitations) should be separated more (possibly by adding vertical line?).

R1a) We have reformatted Table 1 to highlight the headings and better separate the columns.

C1b) UCB: “generally, less contaminated” What does it mean? Does it mean that UCB are never contaminated with EBV or CMV? It should be explained better, at least in the text.

R1b) We have updated the wording of Table 1 to better explain our meaning in this case. Our intention was to highlight that reduced T cell contamination in UCB NK cell preparations reduces the risk of complications like graft vs. host disease. This point is now clearly stated in the table.

C1c) Feeder cells: “Studies have shown effective expansion…” The first part of the sentence (Studies have shown) is not necessary and can be deleted. There is also the number 1 after the ref 177 to be deleted.

R1c) This is a helpful suggestion. We have adjusted the wording and citation accordingly.

C2) I suggest making some minor changes to the figures that already provide a clear explanation of the topic discussed:

C2a) Figure 1 TriKes, as reported in lines 157-158, “have also been modified to incorporate cytokines like IL-15”. It is better to represent the cytokine differently from the antibody fragments.

R2a) We have edited Figure 1 to reflect these comments.

C2b) Figures 2 and 3 are not mentioned in the text. Please add the cross-reference in the text.

R2b) We have added references to Figure 2 (lines 194, 274, 299, and 356) and Figure 3 (lines 284, 348, 366) to the text of the revised manuscript.

C2c) Figure 3: I suggest moving the panels to follow the order of the different paragraphs (TGF-beta, hypoxia, and IDO). The addition of a letter to indicate the different panels allows to mention of Figure 3 and the appropriate panel in the text.

R2c) This is a helpful suggestion. The figure has been reformatted accordingly.

C3) A good summary of the current and updated literature is provided even though some references are lacking, not fully appropriate, or incorrectly cited: 

- Line 44: Reference 5 (Roda, D.; Jimenez, B.; Banerji, U. Are Doses and Schedules of Small-Molecule Targeted Anticancer Drugs Recommended by Phase I Studies Realistic? Clinical Cancer Research 2016) is not adequate for cytokine and chemokine production by NK cells.

- Line 53: it would be better to include a more general reference on ADCC by NK cells (and other NK cell functions); for instance, Vivier E. et al. Nat Immunol. 2008.

- Line 71: there are different papers on IL-15 superagonist ALT803 (or N803) more suitable than ref 21. One is “First-in-human phase 1 clinical study of the IL-15 superagonist complex ALT-803 to treat relapse after transplantation by Romee R. et al Blood, 2008”, but there are also many others.

- Line 147: references 63 and 64 concern CAR-T cells. Please provide adequate references for “…the use of CAR-NK cells for the treatment of non-malignant targets, such as HIV-infected cells.” In addition, in the context of CAR-NKs, the work of K. Rezvani's group is extremely relevant and should be mentioned (for example Liu et al. N. Engl. J. Med., 2020).

- Lines 357-358: The sentence “In addition to HIV (?)…” is not clear. References 55, 143-145 are not appropriate.

The whole manuscript should be carefully checked for missing or inappropriate citations.

R3) We thank the reviewer for the careful review of our citations. We have updated the references to reflect these comments. We have also conducted a thorough review to ensure all references are appropriate to the text.

C4) Paragraph 2.4: In addition to BiKes and TriKes the NK cell engager platform (ANKET), which represents a technological platform for harnessing NK cells as next-generation cancer immunotherapies, should also be mentioned and appropriate references included. These NK cell engager molecules trigger NK cell-activating receptors as NKp46 (or NKp30) together with CD16a and target a tumor-associated antigen (TAA) (Gauthier L. et al. Cell, 2019; Colomar-Carando et al. Cancer Immunol Res, 2022; Gauthier L et al. Nat. Biotechnol. 2023). A tetraspecific CD20-ANKETs (IL-2v/aNKp46/Fc/aCD20) which also contains an IL-2v peptide stimulating IL-2R without CD25 involvement has been also produced (De Maria O. et al. Cell Reports Medicine, 2022) and can be mentioned.

R4) We thank the reviewer for providing the suggestion on ANKETs. This information has been added to Section 2.4 of the revised manuscript.

Reviewer 2 Report

While this review is easy to read, it covers a lot of information that is already well known in the field.  The novel aspect, NK plasticity, which was highlighted in the title and abstract should have been reviewed in greater depth.

Author Response

Response to Reviewer 2

C1: While this review is easy to read, it covers a lot of information that is already well known in the field. The novel aspect, NK plasticity, which was highlighted in the title and abstract should have been reviewed in greater depth.

R1: We thank the reviewer for their positive assessment of our manuscript. We have added an additional section on NK cell plasticity in what is now Section 3.1 of the revised manuscript (line 205). The section complements the elements of plasticity covered in our section on the immunosuppressive actions of TGFb. The section also highlights the original reports of NK cell plasticity and the impact of phenotype conversion on the capacity of NK cells to fight tumor growth and metastasis.

Reviewer 3 Report

In this interesting review, the authors summarized the current understanding of NK cell based adoptive cell therapy and the main obstacles in tumor microenvironment. Besides carefully introduced various NK cell therapies, authors also take a closer look at three immune suppressive factors: hypoxia, TGFb and IDO.  The linguistic style of the manuscript is very good, making the review easy to read and understand. This work has practical worth and will find the interest of your readers. Therefore, I strongly support publication.

Major:

The authors give a comprehensive review of current NK cell therapy. A specific section for future perspective based on the knowledge will strengthen the publication.

The three suppressive factors in TME could lead to NK cells dysfunction in different ways or they target the same manner (for example, immunometabolism), would this be reversible or not?  This would correspond to the title that how we could address this NK cell dysfunction problem, or at least the directions we should try.

Minor:

Line 51 should cite the original paper for the “missing self” K Kärre.

Please carefully check the reference style and font size.

Author Response

Response to Reviewer 3

In this interesting review, the authors summarized the current understanding of NK cell based adoptive cell therapy and the main obstacles in tumor microenvironment. Besides carefully introduced various NK cell therapies, authors also take a closer look at three immune suppressive factors: hypoxia, TGFb and IDO.  The linguistic style of the manuscript is very good, making the review easy to read and understand. This work has practical worth and will find the interest of your readers. Therefore, I strongly support publication.

We thank the reviewer for their positive assessment of our manuscript.

Major:

C1) The authors give a comprehensive review of current NK cell therapy. A specific section for future perspective based on the knowledge will strengthen the publication.

R1) We thank the reviewer for their positive assessment of our manuscript. We have modified our conclusions paragraph, which is now entitled “Conclusions and Future Perspectives” to have a greater emphasis on the future application of enhanced NK cell-based therapies, both for cancer and other diseases of impaired NK cell function.

C2) The three suppressive factors in TME could lead to NK cells dysfunction in different ways or they target the same manner (for example, immunometabolism), would this be reversible or not?  This would correspond to the title that how we could address this NK cell dysfunction problem, or at least the directions we should try.

R2) We thank the reviewer for this insightful suggestion. In the revised manuscript, we have revised our section on NK cell plasticity to include a new section on this topic (line 205) and incorporated discussion of the effects of both TGFb (line 243) and hypoxia (line 303) on NK cell bioenergetic metabolism. Immunometabolism as a promising target to address both TGFb and hypoxia-mediated NK cell suppression is also addressed on line 406 of the revised conclusion.

Minor:

C3) Line 51 should cite the original paper for the “missing self” K Kärre.

R3) We have modified the citation accordingly.

C4) Please carefully check the reference style and font size.

R4) We noted significant issues with font and reference style in the original manuscript. These points have now been addressed.